# StructDiffusion: Object-Centric Diffusion
# for Semantic Rearrangement of Novel Objects

Weiyu Liu[1], Tucker Hermans[2], Sonia Chernova[1], Chris Paxton[3]

*"Make a small circle out of these objects in the center of the table"*

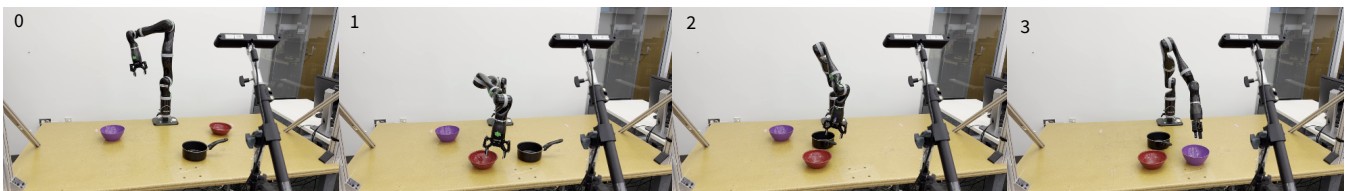

*"Make a tower in the center of the table"*

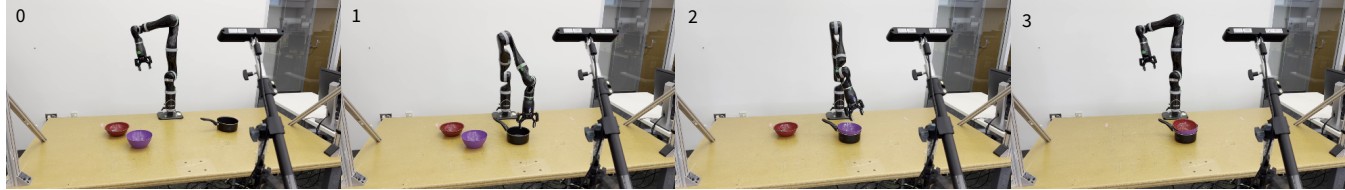

*"Make a short line out of these objects in the center of the table"*

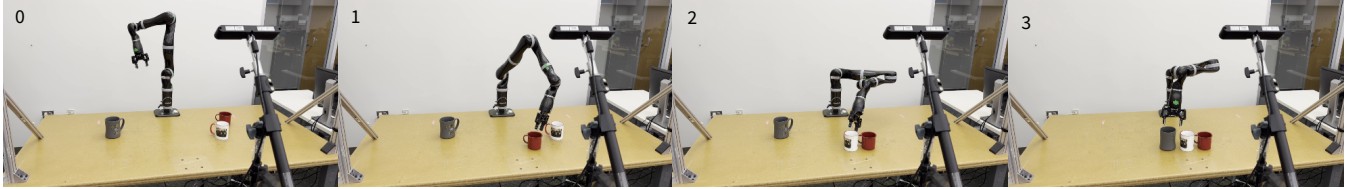

**Fig. 1:** Real-world rearrangement with unknown objects, given a language instruction. We use StructDiffusion to predict possible goals, which we can refine in order to satisfy physical constraints such as avoiding collisions between objects. StructDiffusion is based on an object-centric multimodal transformer backbone, and uses a method based on diffusion models to generate its high-level motion plan.

*Abstract*— **Robots operating in human environments must be able to rearrange objects into semantically-meaningful configurations, even if these objects are previously unseen. In this work, we focus on the problem of building physically-valid structures without step-by-step instructions. We propose StructDiffusion, which combines a diffusion model and an object-centric transformer to construct structures out of a single RGB-D image based on high-level language goals, such as "set the table". Our method shows how diffusion models can be used for complex multi-step 3D planning tasks. StructDiffusion improves success rate on assembling physically-valid structures out of unseen objects by on average $16\%$ over an existing multimodal transformer model, while allowing us to use one multi-task model to produce a wider range of different structures. We show experiments on held-out objects in both simulation and on real-world rearrangement tasks. For videos and additional results, check out our website: http://weiyuliu.com/StructDiffusion/.**

## I. INTRODUCTION

For robots to be successful assistants and collaborators, they must understand object structures. Structures are everywhere in the real world: shelves are stocked, tables set, furniture assembled. Capturing all of these relationships requires models that can reason over both object geometry

and semantics at once, and also reason about physical validity when dealing with objects that have never been seen before.

Building semantically meaningful structures with unseen objects requires satisfying two different sets of constraints: first, we must place objects in the correct positions to satisfy the desired spatio-semantic relations; second, we must ensure that objects are not colliding and arrangements are structurally sound. We propose *StructDiffusion*, a single framework that jointly optimizes over these two, sometimes contrasting, constraints. We hypothesize that by iteratively refining predicted goals subject to these learned constraints, we can better scale and generalize than if we just directly regress to a single solution from a language-conditioned model. We handle these constraints in two ways: first, we train a *language-conditioned object-centric diffusion* model from which we can simultaneously sample goal poses for multiple objects; and second, we train a *discriminator* model that looks at the imagined scenes to reject unrealistic samples. Compared to a language-conditioned multi-task model, *StructDiffusion* achieves a $13.9\%$ higher success rate on previously unseen objects.

Previous work in language-conditioned rearrangement has looked at 2D image representations [1], [2] instead of 3D ones. Other prior work which can perform some rearrange-

[1]Georgia Tech, [2]University of Utah and NVIDIA, [3]Meta AI

ment tasks in 3D [3], [4] have limited generalization to novel objects. Others, like StructFormer [5], regress to only a single solution, and cannot handle multiple challenging types of structures in a single model. To our knowledge, *StructDiffusion* is the first work that combines a multi-modal transformer with a diffusion model to generalize over different objects given high-level language instructions; our method also overcomes limitations of prior techniques regarding voxelization and workspace size [1], [2], [3].

In our approach, we use unknown object instance segmentation to break our scene up into objects, as per prior work [6], [7], [8], [9]. Then, we use a multi-modal transformer to combine both word tokens and object encodings from Point Cloud Transformer [10] in order to make 6-DoF goal pose predictions. These predictions are both refined iteratively via diffusion and selected with a "discriminator" model that learns to recognize unrealistic samples. Our planning-inspired approach is based on three key ideas:

1) An object-centric transformer learns how to construct different types of multi-object structures from observations of novel objects and language instructions;
2) A diffusion model captures a diverse distribution of semantic structures, useful for refinement and planning;
3) A learned discriminator model which dramatically improves performance by rejecting samples violating physical and structural constraints.

To summarize, we propose the first diffusion-based planner for rearrangement of previously-unseen objects, and demonstrate that it generates realistic samples better than baselines in both the real world and simulation.

## II. RELATED WORK

Language-conditioned robot skill learning is a growing area of research [1], [11], [5], [3], [12]. Recently. Say-Can [13] showed how a large language model (LLM) can be used to sequence robot skills to respond to a wide range of natural-language queries. This has been extended to use a map and object-centric representation of the world [14]. A set of other works have used language to build various complex structures [15], [7], [16], [17].

One thread of work looks at language-conditioned skills, but these must themselves be sequenced in order to create more complex structures or behaviors [1], [12], [3]. Another looks at a purely language-based version of the world [7], [15], [13]. ProgPrompt [7] and Code-As-Policies [15] require lots of prompt engineering and object detection capabilities, but they can complete fairly complex structures. Several of these works use an object-centric representation of the world [2], [14], [5], where objects are segmented or detected and encoded separately. For example, VIMA [2] used encoded object patches as input to a multimodal transformer. None of these works, however, look specifically at how we can ensure we are generating *physically realistic* structures: in our experiments, we show how these direct-regression-first approaches do not generate the same quality of structures, and that in particular, simply predicting placement poses or actions will lead to more failures.

Several works have looked at planning with unknown objects. Simeonov et al. [18] propose a planning framework for rigid body objects, but they do not create structures with complex dependencies. Curtis et al. [19] investigate task and motion planning with unknown objects, which relies on a similar segmentation and grasping pipeline to our work, but does not look at learning goals, instead assuming access to a set of predicates which can be evaluated at planning time.

Finally, there is a set of works which learn object-object relations for planning [6], [20], [21], which is relevant to our method's refinement process. Many of these do so explicitly. In particular, learning object skill preconditions is very useful for sequential manipulation, so some works look at predicting relationships in this context [22], [21], [23]. For example, SORNet [21] learns to predict relations between objects given a canonical image view of the objects; similarly a predictive model from image inputs is learned for capturing relationships [23]. These kinds of relations are an important part of planning sequential manipulation as per *StructDiffusion*, but we look only at implicitly classifying object relationships as a whole.

Closely related to our use of diffusion model is DALL-E-Bot [17], which uses DALL-E to generate a goal image for an arrangement of multiple objects, then uses object matching to rearrange these objects. This method makes a common assumption [2], [1] which is that there will be an unoccluded top-down view of the scene. This assumption is necessary for their method because they use a combination of image captioning and object recognition to convert between a diffusion image and a motion plan. In our case, we are not restricted by which position or angle the image is generated from; we can directly refine object placement poses given point cloud observations.

## III. PRELIMINARIES

We provide background information on diffusion models [24], [25] and transformers [26]. These two neural network structures provide core components for our approach.

### A. Diffusion Models

Denoising Diffusion Models are a class of generative models [24], [25]. Given a sample $x \sim q(x_0)$ from the data distribution. The *forward* diffusion process is a Markov chain that creates latent variables $x_1, ..., x_T$ by gradually adding Gaussian noise to the sample:

$$q(x_t|x_{t-1}) = \mathcal{N}(x_t; \sqrt{1 - \beta_t}x_{t-1}, \beta_t \mathcal{I})$$

Here $\beta_t$ follows a fixed variance schedule such that the variance at each step is small and the total noise added to the original sample in the chain is large. These two conditions allows sampling $x_0 \sim p_\theta(x_0)$ from a *reverse* process that starts with a Gaussian noise $x_T$ and follows a learned Gaussian posterior

$$p_\theta(x_{t-1}|x_t) \sim \mathcal{N}(x_{t-1}; \mu_\theta(x_t, t), \Sigma_\theta(x_t, t))$$

In this work, we adopt the simplified model introduced in [27] that fixes the covariance $\Sigma_\theta(x_t, t)$ to an untrained time-dependent constant and reparameterize the mean $\mu_\theta(x_t, t)$

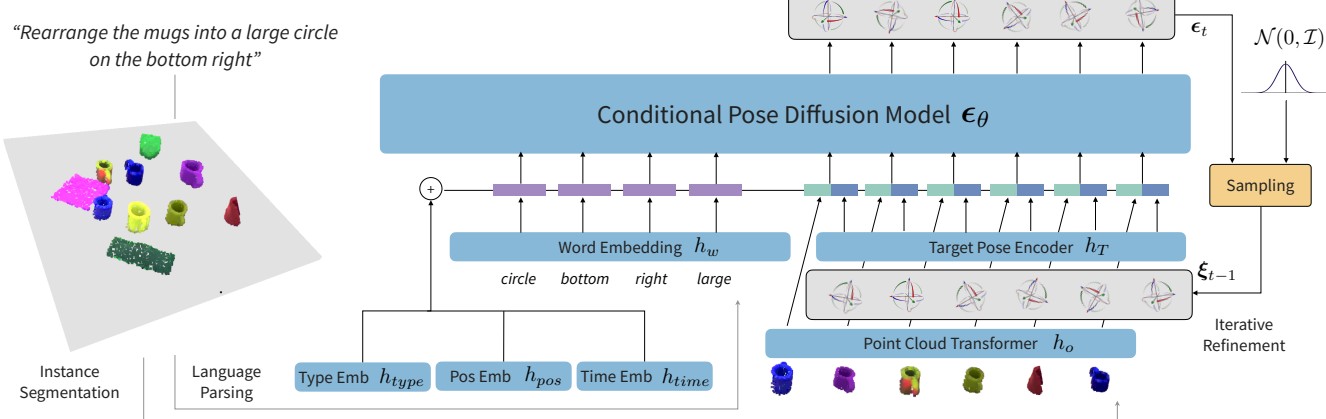

*"Rearrange the mugs into a large circle on the bottom right"*

**Fig. 2:** Overview of the object-centric diffusion model. We combine a diffusion model with an object-centric multimodal transformer to iteratively reason about both 3D object embeddings and task specification, and predict goal poses of objects.

with a noise term $\epsilon_t$. Diffusion models can be trained to minimize the variational lower bound on the negative log-likelihood $\mathbb{E}[-\log p_\theta(x_0)]$. A simplied training objective with the reparameterized mean can be derived as:

$$L_{\text{simple}} = \mathbb{E}_{t \sim [1,T], x_0 \sim q(x_0), \epsilon \sim \mathcal{N}(0,\mathcal{I})}[||\epsilon - \epsilon_\theta(x_t, t)||^2]$$

Diffusion models have been used for motion and grasp planning in prior work [28], [29]. However, existing methods require known object models and are not conditioned on flexible language goals.

*B. Transformers*

Transformers were proposed in [26] for modeling sequential data. At the heart of the Transformer architecture is the scaled dot-product attention function, which allows elements in a sequence to attend to other elements. Specifically, an attention function takes in an input sequence $\{x_1, ..., x_n\}$ and outputs a sequence of the same length $\{y_1, ..., y_n\}$. Each input $x_i$ is linearly projected to a query $q_i$, key $k_i$, and value $v_i$. The output $y_i$ is computed as a weighted sum of the values, where the weight assigned to each value is based on the compatibility of the query with the corresponding key. In this work, we use the encoder layers in the original transformer architecture. Each encoder layer includes an attention layer and a position-wise fully connected feed forward network. With the use of attention mask, the encoder layer can process sequences with different lengths.

IV. *StructDiffusion* FOR OBJECT REARRANGEMENT

Given a single view of an initial scene containing objects $\{o_1, ..., o_N\}$ and a language specification containing word tokens $\{w_1, ..., w_M\}$, our goal is to rearrange the objects to reach a goal scene that satisfy the language goal. We assume the objects are rigid and we are given a partial-view point cloud of the scene with segment labels for points to identify the objects. We can extract the initial poses of the objects $\{\xi_1^{pc}, ..., \xi_N^{pc}\}$ from the segmented object point clouds $\{x_1, ..., x_N\}$ by setting the rotation to zero and the position to the centroid of each object point cloud in the world frame.

To rearrange the objects, the robot needs to move the objects to their respective goal poses $\xi_i^{goal}$.

In this work, our robot can execute pick and place actions. For each object, we can sample a set of stable grasps $\mathcal{G} = \{g_1, ..., g_M\}$. Given a target pose for object $\xi_i^{goal}$ and a stable grasp $g_j$, the robot can move its end effector to $\xi_i^{ee} = \xi_i^{goal}(\xi_i^{pc})^{-1}g_j$ to place the object at the goal pose. We only use pick and place actions in our setup to simplify the problem. However, our object-centric actions can be integrated with sampling-based TAMP solutions [19] to also leverage other motion primitives, such as pushing and regrasping, to reach the goal poses predicted by our system.

Below we describe our approach to sample goal poses for objects based on partial point clouds of the objects and the language goal. Our framework combines a generator based on a diffusion model and a learned discriminator that filters invalid samples. Our diffusion model is integrated with a transformer model that maintains an individual attention stream for each object. This object-centric approach allows us to focus on learning the interactions between objects based on their geometric features as well as the grounding of abstract concepts on spatio-semantic relations between objects (e.g., large, circle, top). The discriminator model operates on the scene level after transforming the objects to their predicted goal poses to further reject invalid samples.

*A. Encoders*

We leverage modality-specific encoders to convert the multimodal inputs to latent tokens that are later processed by the transformer network.

**Object encoder.** Given the segmented point cloud $x_i$ of an object $o_i$, we learn an encoder $h_o(x_i)$, in order to obtain the latent representation of the object. This is based on Point Cloud Transformer (PCT) [10], which has been shown to be effective at shape classification and part segmentation. We process the centered point cloud with PCT and learn a separate multilayer perceptron (MLP) to encode the mean position of the original point cloud. Encodings from the two networks are concatenated to give $h_o(x_i)$. We rely on this

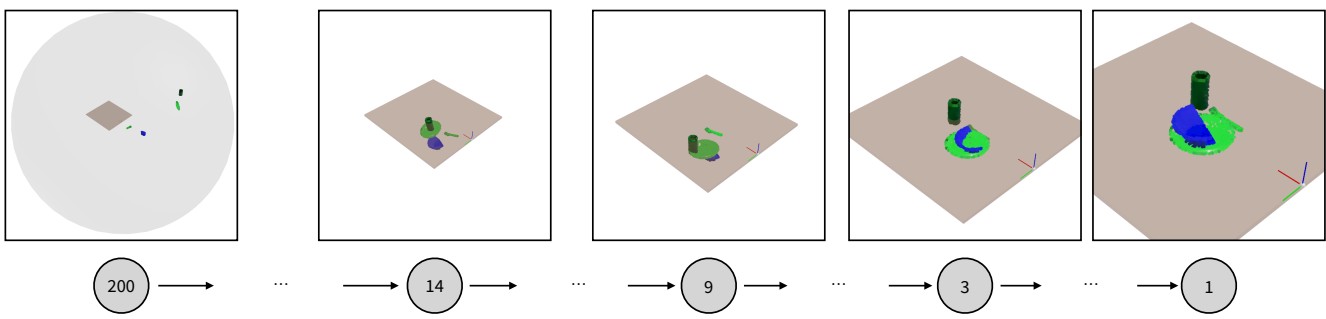

**Fig. 3:** We model high-level structure planning as a diffusion process. We start from the last step of the reverse diffusion process and jointly predict goal poses for all objects in the scene. This formulation allows our model to reason about object-object interactions in a generalizable way, which outperforms simply predicting goal poses from multi-modal inputs.

latent representation of objects for semantic, geometric, and spatial reasoning.

**Language.** To map the language goal to its latent representation, we map each unique word token from the language instructions separately to an embedding with a learned mapping $h_w(w_i)$. This method helps establish a fine-grained correspondence between each part of the language specification and the respective constraint on the generated structure.

**Diffusion encodings.** Since the goal poses of objects are iteratively optimized by the diffusion model and need to feed back to the model, we use a MLP to encode the goal poses of the objects $h_T(\xi_i^{goal})$. To compute the time-dependent Gaussian posterior for reverse diffusion, we combine a latent code for $t$ in the feature channel by learning a time embedding $h_{time}(t)$.

**Positional encoding.** To differentiate the multimodal data, we use a learned position embedding $h_{pos}(i)$ to indicate the position of the words and objects in input sequences and a learned type embedding $h_{type}(v_i)$ to differentiate object point clouds ($v_i = 1$) and word tokens ($v_i = 0$).

### B. Conditional Pose Diffusion Model

Combining a diffusion model and an object-centric transformer, *StructDiffusion* can sample diverse yet realistic object structures while accounting for the complex constraints imposed by the object geometry and language goal. The conditional diffusion model predicts the goal poses for the objects $\boldsymbol{\xi}_0 = \{\xi_i\}_i^N$ starting from the last time step of the reverse diffusion process $\boldsymbol{\xi}_T \sim \mathcal{N}(0, \mathcal{I})$, as illustrated in Fig. 3. We use the bold symbol here because we jointly optimize the poses of all objects.

Different from most existing diffusion models that directly generate goal images and do not explicitly model individual objects [24], [25], [17], we use the transformer model to build an object-centric representation of the scene and reason about the higher-order interactions between multiple objects. This approach allows us to account for both global constraints and local interactions between objects. Leveraging attention masks, a single transformer model can also learn to rearrange different numbers of objects.

The use of the diffusion model helps us capture diverse structures since we are sampling from a series of Gaussian

noises at different scales when going from $\boldsymbol{\xi}_T$ to our goal $\boldsymbol{\xi}_0$. The resulting samples, therefore, is diverse at different levels of granularity (e.g., different placements of the structures and different orientations of the individual objects). The diversity is also crucial when dealing with the inherent ambiguity in language instructions. For example, a *large* circle of plates and a *large* circle of candles impose different constraints on the sizes of the structures because the objects being arranged have different sizes.

Combining the advantages of the object-centric transformer and the diffusion model, we propose to model the conditional reverse process as

$$p_\theta(\boldsymbol{\xi}_0|\{x_i\}, \{w_i\}) = p(\boldsymbol{\xi}_t) \prod p_\theta(\boldsymbol{\xi}_{t-1}|\boldsymbol{\xi}_t, \{x_i\}, \{w_i\})$$

The generation process depends on the point clouds of the objects and language instruction. As discussed in III-A, we learn the time-dependent noise $\epsilon_t$, which can be used to compute $\boldsymbol{\xi}_t$. We use the transformer as the backbone to predict the conditional noise $\epsilon_\theta(\boldsymbol{\xi}_t, t, \{x_i\}, \{w_i\})$ for each object. We obtain the transformer input for the language part and the object part as

$$c_{i,t} = [h_w(x_i); h_{pos}(i); h_{type}(v_i); h_{time}(t)]$$
$$e_{i,t} = [h_o(x_i); h_T(\xi_i^{goal}); h_{pos}(i); h_{type}(v_i); h_{time}(t)]$$

where $[;]$ is the concatenation at the feature dimension. The model takes in the sequence $\{c_{1,t}, .., c_{M,t}, e_{1,t}, ..., e_{N,t}\}$ and predicts $\{\epsilon_{1,t}, ..., \epsilon_{N,t}\}$ for the object poses. We parameterize 6-DoF pose target $\xi$ as $(t, R) \in SE(3)$. We directly predict $t \in \mathbb{R}^3$ and predict two vectors $a, b \in \mathbb{R}^3$, which are used to construct the rotation matrix $R \in SO(3)$ using a Gram–Schmidt-like process proposed in [30].

### C. Discriminators

Besides the generator, we can also use a learned discriminator model to further filter the predictions for realism. The discriminator works on imagined scenes, where the point clouds of objects are rigidly transformed to the respective goal poses following $x_i^{goal} = \xi_i^{goal}(\xi_i^{pc})^{-1}x_i$. Here we also have the opportunity to leverage a spatial abstraction different from the one used by the generator. The generator operates the latent object-centric representation that are suitable to *imagine* possible structures. For a discriminator, the interactions between the transformed point cloud objects

**Algorithm 1** Planning with *StructDiffusion*

---

1: **for** $t \in \text{range}(T, 1)$ **do**
2:      $\epsilon_t \sim \epsilon_\theta(\boldsymbol{\xi}_t, t, \{x_i\}, \{w_i\})$
3:      $z \sim \mathcal{N}(0, \mathcal{I})$ if $t > 1$ else $z = 0$
4:      $\boldsymbol{\xi}_{t-1} = \frac{1}{\sqrt{\beta_t}}(\boldsymbol{\xi}_t - \frac{\beta_t}{\sum_{s=1}^{t} 1 - \beta_t} \epsilon_t) + \sqrt{\beta_t} z$
5: Transform object points: $x_i^{goal} = \xi_i^{goal}(\xi_i^{pc})^{-1} x_i$
6: Compute discriminator scores
7: **return** ranked $\boldsymbol{\xi_1}$

---

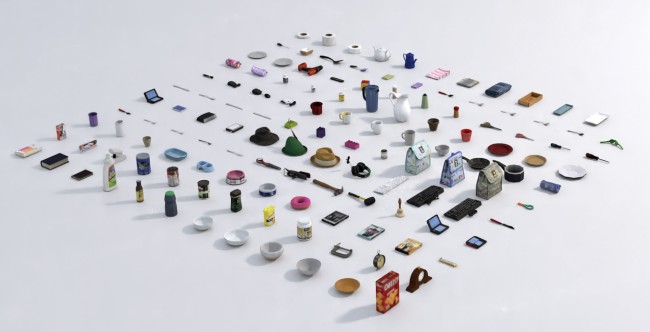

**Fig. 4:** Testing objects from Google Scanned Objects [35], Replica-CAD dataset [36], and YCB Object Set [37]. The test object dataset contains a wide range of textured objects belonging to various classes. None of these objects appear in the training data.

can be directly reasoned at the point level. To maintain the ability to distinguish each individual objects, we add a one-hot encoding to each point feature. In our preliminary experiment, we found that the scene-level collision model has more discrimination power than the object-centric model that operates on latent representation of objects.

We explore two discriminator models. The first collision discriminator is learned to predict pairwise collisions between two objects from their partial point clouds. The second structure discriminator is learned to classify the whole multi-object structure. Similar to the language-conditioned generator, we also condition the structure discriminator such that the discriminator can learn structure-specific constraints to score the samples. We found that the structure discriminator works better when it is only required to predict if local constraints are satisfied. Therefore, we normalize the scene point cloud and drop parts of the language instruction that specify global constraints such as where to place the structure on the table.

### D. Planning and Inference

In Alg. 1, we show how to combine the different components of our framework to sample object structures. We first initialize a batch of goal poses $\mathbb{R}^{\in B \times N \times (3+3+3)}$ with random noise. We use batch operation on a GPU to perform diffusion and transform point clouds of multiple objects for different samples. For the discriminators, we also generate combined point cloud of objects after the diffusion process and score them in batches. The ranked samples are returned. Each sample corresponds to a physically and semantically valid multi-object structure that can be used by other components of the manipulation pipeline for planning.

### E. Training Details

To train our model, we use the dataset from [5] containing tuples $(\{x_i\}, \{T_i^{goal}\})$. We train a single model for all structures where the number of examples for different classes of structures are balanced. We use a batch size of 128 and train the diffusion model on a single RTX3090 GPU for about 12 hours. To train the collision discriminator, we randomly sample $100,000$ configurations of objects. For the structure discriminator, we generate negative examples by randomly perturbing the ground truth target poses $\xi_i^{goal}$. For each negative example, we also randomly select a set of objects to perturb so that there are negative examples that have different number of objects out of place. We augment the data by using point clouds from different time steps of

the rearrangement sequence to create the imagined scene as they usually create different occlusions for objects.

## V. SIMULATION EXPERIMENTS

We first evaluate our method in simulation.

### A. Baselines

- **StructFormer**: This baseline uses a multimodal and object-centric transformer network to generate multi-object structures based on segmented object point clouds and language instructions [5]. The transformer network autoregressively predicts the goal poses of each object. We follow the original work to train a separate model for each class of structure.

- **CVAE**: This baseline is based on a conditional variational autoencoder (CVAE) model. CVAEs have been used to capture the different modes for multi-task learning and language-conditioned manipulation [11], [12]. We introduce a CVAE that uses the object-centric transformer backbone as a strong baseline for semantic rearrangement. To prevent the latent variable being ignored when combining the transformer with CVAE, the transformer network predicts the object goal poses in a single forward pass (i.e., not autoregressively). A single model is trained for four classes of structures.

- **Optimization with Learned Discriminator**: This baseline iteratively optimizes the goal poses of objects with the structure discriminator that is trained to classify valid rearranged scenes and invalid ones. This general approach has been used extensively for learning language-conditioned manipulation from offline data [31], grasping [32], [33], and predicting stable placements of objects [6], but not for language-conditioned multi-object rearrangement. We uses the cross-entropy method for optimization [34]. We only optimize the object poses and not the structure pose to simplify the optimization problem. We initialize the samples from the baseline generative models because initializing the variables with random values does not lead to meaningful performance.

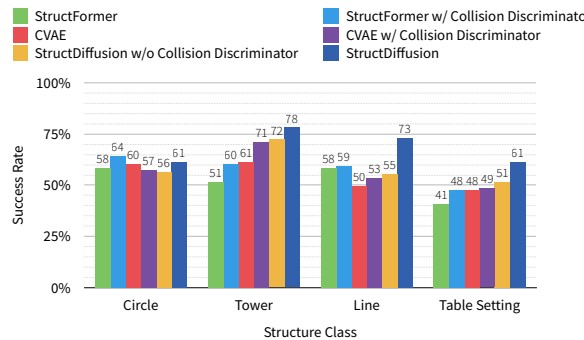

**Fig. 5:** Success rates for four different classes of structures on held-out objects. Models are evaluated in a physics simulator using unseen objects. A rearrangement is successful only if all objects are placed in physically valid poses and the rearranged scene satisfies the language goal. Compared to StructFormer [5], the model previously proposed for semantic rearrangement, *StructDiffusion* obtains a 16% average improvement in success rate.

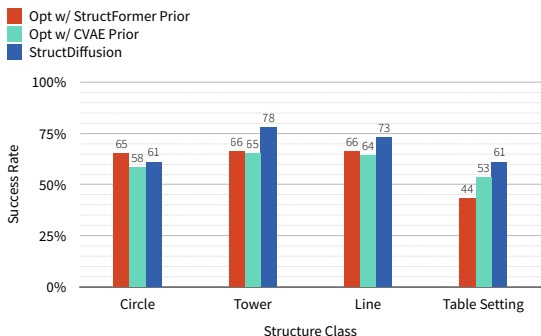

**Fig. 6:** Comparing *StructDiffusion* with other iterative methods. The two baselines initialize samples of target object poses using either the StructFormer or the CVAE model. The predicted scores of a learned discriminator is then used to guide iterative optimization of the samples. In comparison, *StructDiffusion* directly predicts the noises $\epsilon_t$ that need to be removed from the samples at each step.

### B. Experimental Setup

We evaluate all models in the PyBullet physics simulator [38]. Point cloud observations are rendered with NViSII [39]. We test on novel object models from both known and unknown categories as our goal is to transfer the model learned in simulation directly to real-world objects. Fig. 4 shows the testing objects, which are collected from Google Scanned Objects [35], ReplicaCAD dataset [36], and YCB object Set [37]. To generate the test scenes, we use the same data collection pipeline that is used to collect groundtruth data from prior work [5]. This ensures that a valid rearrangement can be found for each scene. The set of objects and the language goal for each scene are randomly sampled. Distractor objects are randomly placed in the scene to simulate occlusions.

We report success rate for the rearrangements. To isolate the pose prediction problem from other components of the system (e.g., grasp sampling and motion planning), we directly place objects 3cm above the the predicted target poses. We checks whether the rearrangement is physically valid by running the simulation loop after placing each object. We check possible collisions and intersections between objects using approximate convex decompositions of the 3D object models. We also implement model-based classifiers to evaluate whether the rearrangement satisfy the language goal. For example, we check whether the objects are in a line using the centroids of the models. A rearrangement is considered as successful if the placements of objects are not preempted due to physics-related failures and the goal scene satisfies all semantic constraints determined by the given language goal. On average, there are 5 constraints for different types of structures.

### C. Comparison with Other Generative Models

In Fig. 5, we compare with other generative models and gain insights into the generator-discriminator design of our model. We see that our complete model, *StructDiffusion*, drastically outperformed all baselines on the tower, line,

and table setting structures and obtained comparable performance on the circle structures. The improvement was most significant for structures that required precise placements of objects and modeling contacts between objects. The generator-discriminator design was necessary because the diffusion model alone still generated invalid samples, especially for the line structures. The performance difference between *StructDiffusion* and the ablated model that does not use discriminator supports that our model can leverage the complimentary strengths of object-centric representation and scene-level representation that preserves the point-to-point interactions. Although applying the collision discriminator also improved the performance of StructFormer and CVAE, our diffusion model benefited the most from the addition. We attributes this difference to the different diversities of samples from these three classes of generative models. The autoregressive transformer underlying StructFormer does not explicitly model uncertainty, therefore leads to similar samples for each scene. The single source of stochasticity from the latent variable of the CVAE model is also not enough. As the diffusion model incorporates uncertainties at different scales, it has the ability to both generate different classes of structures but also generate hypotheses of object placements given only partial, and even heavily occluded, point cloud of objects. We provide qualitative comparison in Fig. 7.

### D. Comparison with Other Iterative Methods

In Fig. 6, we compare *StructDiffusion* with other optimization-based baselines that can take advantage of the additional computational time to iteratively refine the prediction. The result shows that *StructDiffusion* outperformed the other two baselines. Even though strong performance was observed when applying optimization-based method to other manipulation tasks, we do not see significant benefit in our task. Looking more closely, we observe that the challenging cases that are not yet solved by the non-iterative variants are cases where the placements of objects are closely related (e.g., mugs tightly packed in a line without intersection, the third task in Fig. 7). In these cases, the

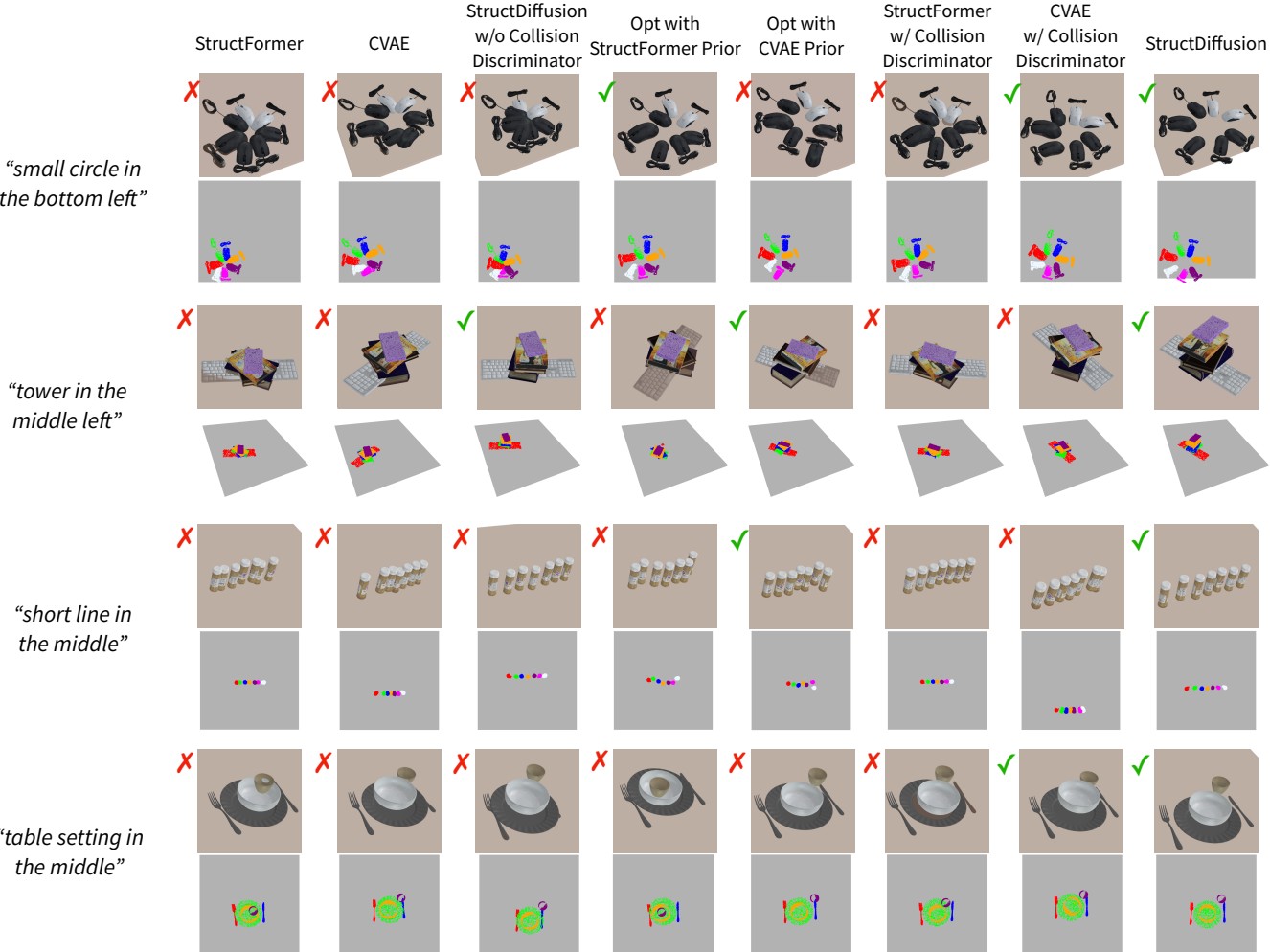

**Fig. 7:** Comparison between *StructDiffusion* and the baselines on partial views of held-out objects, given language commands from four different categories. *StructDiffusion* is better at resolving constraints involving contact and precise arrangement of objects, avoiding collisions and creating physically realistic placements. The labels indicate whether the structures can be successfully built in the simulation environment and also satisfy the language goal.

guidance from the discriminator can be ambiguous and leads to local minimal without reaching valid solutions. We hypothesize the leveraging guidance at different scales is necessary, as studied in a recent work that directly learns to predict scores (i.e., gradients) at different scales for 2D object rearrangement [40]. Score-based method is closely related to the diffusion model we used in this work, as shown by Song and colleagues [25].

## VI. REAL WORLD EXPERIMENTS

We also performed a set of real world experiments on real data, including testing structure assembly on a robotic manipulation task.

### A. Perception and Hardware

We deployed our system on a 7-DoF JACO arm with an Asus Xtion RGB-D Camera. We obtained segmented object point clouds by identifying clusters-of-interest through table surface detection and Euclidean distance clustering, using the Point Cloud Library [41]. We calculated antipodal grasps over each object point cloud [42], which are then ordered and executed using pairwise ranking [43]. We used RRT-Connect [44] for motion planning. We released each object 3cm above the predicated pose.

### B. Predictions for Real-World Objects

We show examples of the predicted structures for real-world objects in Fig. 8. These examples are created by rigidly transforming the segmented object point clouds from an initial scene with the target poses of the highest ranked structure. Even though our model is trained only on simulation data, it can be directly used to generate semantically diverse and physically valid structures for real-world objects. our model can generate different variations of the same structure type, as shown in *(A, B)*. The same set of objects can be arranged into completely different classes of structures conditioning on the language, as shown in *(A, C)* and *(D, E)*. Besides changing the positions and sizes of the structures, the orientations of the structures can also be specified in language *(F, G)* and *(H, I)*. Note that even

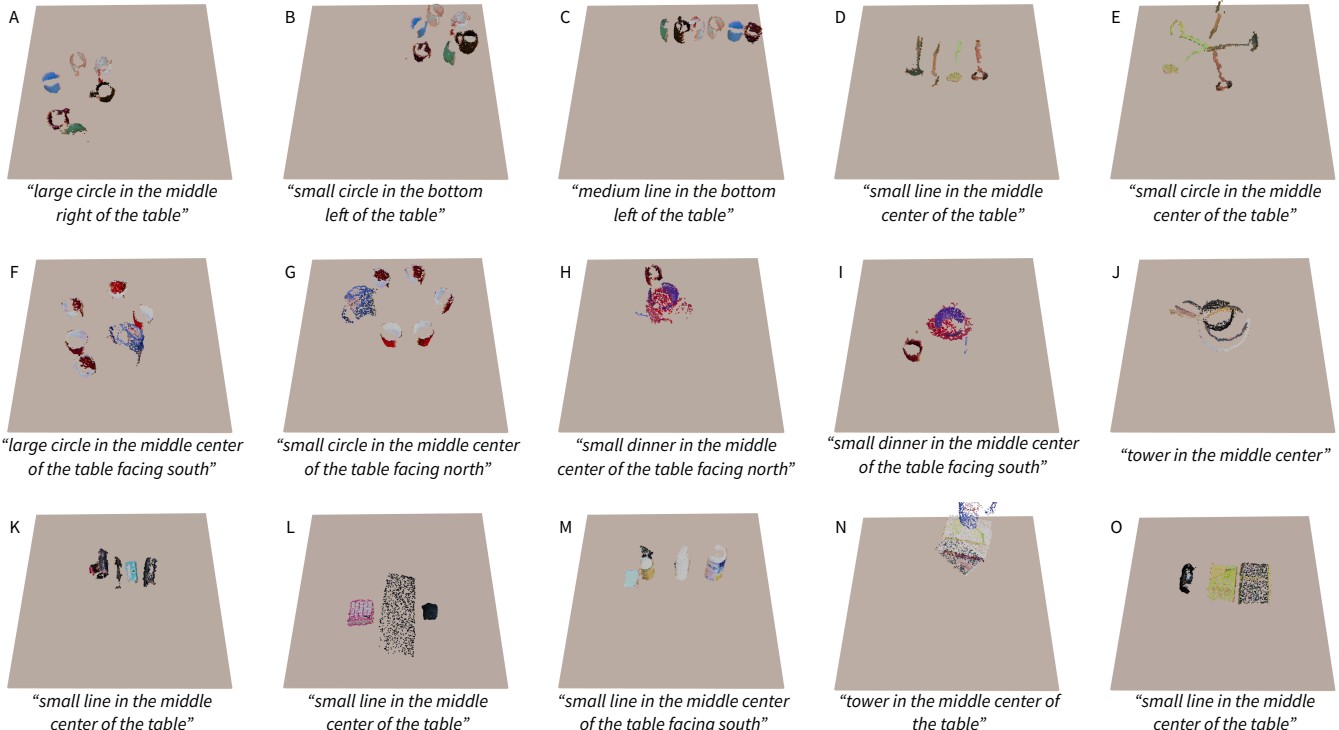

**Fig. 8:** Examples of predicted structures for real-world objects. We can predict structures from raw point clouds for a wide range of language instructions fitting into four different broad classes.

**TABLE I:** Robot experiments with real-world objects. We perform each of the task 3 times with different initial positions of objects. We show the number of times that valid grasp and motion plans are found and that the plans are executed successfully by the robot.

| Objects | Structure | Grasp and Motion Planning | Placement |
|---|---|---|---|
| Bowl, Bowl, Pan | Tower | 3 | 2 |
| Bowl, Bowl, Pan | Small line | 2 | 2 |
| Bowl, Bowl, Pan | Small Circle | 3 | 3 |
| Overall Success Rate | | 88.9% | 77.8% |

though table settings in the training data are only aligned horizontally as shown in *I*, the use of language and training on other orientation-specific structures enable compositional generalization to a new orientation shown in *H*. Finally, we see non-symmetrical object (e.g., mugs, knifes, and spatulas) are correctly aligned in *B, D, E, J, H*.

### C. Rearrangement

To reliably rearrange multiple objects, we combined *StructDiffusion* with grasp and motion planning. We performed nested search to find the target structure to execute. Specifically, we iterate through the generated and ranked structures. For each structure, we sample a set of grasp poses for each object and compute corresponding pre-grasp, standoff, and placement poses based on the prediction. We searched for valid motion plans between these waypoints. If all motion plans have been found, we execute on the robot.

In Table I, we show success counts and average success rate for trials with different objects and different language goals. Valid motion and grasp plans can be found most of the time due to the diverse structures generated by *StructDiffusion*. We observed that partial point clouds due to noisy real sensor and self-occlusions for large objects led to a small number of invalid structure predictions. While planning, we make the assumption that the objects are rigidly attached to the gripper after grasping without slippage. This assumption generally did not hold in the real world and led to occasional failures. This assumption can be relaxed by predicting a post-grasp displacement, using learned models such as [45].

### VII. CONCLUSIONS

We described *StructDiffusion*, an approach for creating physically-valid structures using multimodal transformers and diffusion models. *StructDiffusion* operates on point cloud images of previously-unseen objects, and can create structures for a range of language instructions.

Specifically, we compared to a number of baselines, including the previous state of the art [5] and to a conditional variational autoencoder. End-to-end policies do not perform as well, because they cannot refine placement poses that are *nearly* correct. Using diffusion models for sampling and a trained discriminator to refine object poses significantly improves performance.

In this work, we did not look at optimally planning. In the future, we could look at combining this approach with task and motion planning for unknown objects, as in [19]. In addition, in the future, we could apply our work to a wider range of structures.

## ACKNOWLEDGMENT

We thank Yilun Du for discussions about the use of diffusion models.

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
