# OpenReview forum: "StructDiffusion: Object-Centric Diffusion for Semantic Rearrangement of Novel Objects"
_robot-learning.org/CoRL/2022/Workshop/LangRob — LangRob 2022 Poster_

### Official Review · Reviewer_DB4F · 2022-11-12
**Well presented and novel combination of object-centric representations with diffusion models for 3D planning**

**Rating:** 8
**Confidence:** 3

**Review:**

**Summary**:

This work introduces StructDiffusion, a method that uses a diffusion model and an object-centric transformer to propose multi-step 3D plans. The method proposes two parts: a language-conditioned object-centric diffusion model to satisfy desired goal semantics, and a learned discriminator to reject invalid proposals. They demonstrate that StructDiffusion is able to generalize to novel objects and language instructions to generate useful 3D predictions in both simulation and the real world.


**Strengths**:
- The paper is very well presented and clear to understand
- The combination of diffusion models and object-centric representations for language-conditioned 3D plans is novel

**Weaknesses**:
- It would be useful to see the performance of baseline methods in the real world experiments.

---

### Official Review · Reviewer_QRVu · 2022-11-13

**Rating:** 7
**Confidence:** 3

**Review:**

# Summary of the paper

The authors introduce StructDiffusion, an architecture which combines an object-based transformer with an iterative application of a diffusion model, and a learned discriminator to refine and generate proposed goals (arrangements of objects) which satisfy natural language instructions. They demonstrate an improvement over the previous state of the art, StructFormer, on three of the four tasks considered, and additionally demonstrate a high success rate on a real robot when using the automatically generated goal.

# Strengths and weaknesses
Strengths:
- The paper was well organized, and easy to read, with only a few minor cosmetic errors
- Diffusion models are a recent breakthrough in high definition image generation, so it is valuable to see how they can be used in robotics
- The numerical performance of the model is better than alternatives on several different tasks
- The problem tackled is of significant value, and may be somewhat understudied
- The real robot demonstration shows that the generated goals are usable
- The ablation of the learned classifier was informative
Weaknesses / areas of improvement:
- More than 3 robot trials per task would have been appreciated.
- An analysis of the failure modes of the model would have been nice. Why did StructFormer do better on the circle task?
- In some of the tasks, it was a bit hard to tell the difference between the failing and passing arrangements (especially for line), and in some cases, the failing ones looked better than the passing ones!
- There seemed to be a somewhat nonstandard use of the word "planning" being used in a few places in the paper, without explanation. Is "structure planning" a term of art coined by this paper, or was it previously used? If coined, it could use a proper definition. Otherwise, a citation may be necessary. Same for "planning-inspired". This could use more exposition or should potentially be omitted.

# Quality, clarity, originality, and significance
I find the paper to be of high quality, very clear, and moderately original (a combination of known things in new context). I find the results to be significant.

---

### Decision · Program_Chairs · 2022-11-15

Accept (Poster)